# Cytomegalovirus Infection Impairs the Mobilization of Tissue-Resident Innate Lymphoid Cells into the Peripheral Blood Compartment in Response to Acute Exercise

**DOI:** 10.3390/v13081535

**Published:** 2021-08-03

**Authors:** Eunhan Cho, Bailey Theall, James Stampley, Joshua Granger, Neil M. Johannsen, Brian A. Irving, Guillaume Spielmann

**Affiliations:** 1School of Kinesiology, Louisiana State University, Baton Rouge, LA 70803, USA; echo3@lsu.edu (E.C.); btheal1@lsu.edu (B.T.); jstamp5@lsu.edu (J.S.); jgrang6@lsu.edu (J.G.); njohan1@lsu.edu (N.M.J.); brianairving@lsu.edu (B.A.I.); 2Pennington Biomedical Research Center, Baton Rouge, LA 70808, USA

**Keywords:** cytomegalovirus, innate lymphoid cells, acute exercise, mobilization

## Abstract

Circulating immune cell numbers and phenotypes are impacted by high-intensity acute bouts of exercise and infection history with the latent herpesviruses cytomegalovirus (CMV). In particular, CMV infection history impairs the exercise-induced mobilization of cytotoxic innate lymphoid cells 1 (ILC1) cells, also known as NK cells, in the blood. However, it remains unknown whether exercise and CMV infection modulate the mobilization of traditionally tissue-resident non-cytotoxic ILCs into the peripheral blood compartment. To address this question, 22 healthy individuals with or without CMV (20–35 years—45% CMV^pos^) completed 30 min of cycling at 70% VO_2_ max, and detailed phenotypic analysis of circulating ILCs was performed at rest and immediately post-exercise. We show for the first time that a bout of high-intensity exercise is associated with an influx of ILCs that are traditionally regarded as tissue-resident. In addition, this is the first study to highlight that latent CMV infection blunts the exercise-response of total ILCs and progenitor ILCs (ILCPs). These promising data suggest that acute exercise facilitates the circulation of certain ILC subsets, further advocating for the improvements in health seen with exercise by enhancing cellular mobilization and immunosurveillance, while also highlighting the indirect deleterious effects of CMV infection in healthy adults.

## 1. Introduction

Acute exercise is a robust physiological stressor that promotes lymphocyte mobilization into the circulation [1,2]. This transient increase in the lymphocyte number in peripheral blood post-exercise is due to the exercise-induced stimulation of the sympathetic nervous system [1,3] and the associated increase in cardiac output [1,3], hemodynamic shear stress [1,3], and catecholamine concentration [1,3], all of which are factors leading to tissue-resident lymphocyte mobilization and lymphocyte demargination [4]. This highly conserved response amongst individuals affects various circulating immune cells, including T cells [5], B cells [6], mucosal-associated invariant T cells (MAIT cells) [7], innate lymphoid cells (ILC), and cytotoxic NK cells [8]. Since this exercise-induced cellular mobilization appears to preferentially affect highly cytotoxic and antigen-experienced cells (i.e., NK cells, CD8 T-cells), it has been suggested to play a significant role in exercise-mediated health improvements by promoting immune surveillance [2,3,4]. Interestingly, however, less is known on the effects of acute exercise on the mobilization of traditionally tissue-resident immune cells, especially non-NK-cell ILCs.

ILCs are a highly heterogeneous population of innate immune cells, found across lymphoid and non-lymphoid tissues, including the liver, spleen, lymph nodes, and skin [9,10]. ILCs originate from innate lymphoid cell precursors (ILCPs), which are highly plastic multipotent and unipotent cells that differentiate into ILC subsets depending on microenvironmental cues [10,11]. ILCs play a critical role in tissue homeostasis, host defense against infection, and allergic inflammation, while maintaining adequate communication between the innate and adaptive immune systems [9]. Although ILCs lack antigen receptors, they can be further classified in distinct subsets, coined ILC1, ILC2, and ILC3, based on their effector profiles, cytokine secretion, and differential surface protein expression [9,12]. Specifically, group 1 of ILCs (ILC1) contains conventional cytotoxic NK cells and non-cytotoxic helper ILC1 cells, with both producing high levels of interferon-γ (IFNγ) [9,10]. Group 2 ILCs (ILC2) protect against bacterial challenges and allergens via a cytokine release, which is more traditionally associated with Th2 cells, thus playing a crucial role in the host response to tissue damage and viral infection [12,13]. Group 3 of ILCs (ILC3) is a heterogeneous group that includes lymphoid tissue inducer (LTi) cells and ILC3 cells, which contributes to maintaining the integrity of the intestinal barrier, and protects against microbial pathogens [9,14].

Over the years, the response of cytotoxic ILCs (also known as NK cells) to acute and chronic exercise has been investigated [15,16]. In humans, NK cells are composed of two distinct subsets based on the CD56 expression level: CD56^dim^ and CD56^bright^. Although CD56^dim^ cells are mainly distributed in the peripheral blood, bone marrow, spleen, and lung, CD56^bright^ cells are more widely distributed in the secondary lymphoid organs [17]. Although the overall localization of CD56^dim^ and CD56^bright^ cells is different within individuals, sex and latent cytomegalovirus infection (CMV) status do not appear to affect their distribution at rest [17]. However, their response to acute physiological stressors such as exercise is likely impacted by an individual’s infection history. For example, NK cell mobilization in participants with CMV is blunted, compared to participants without CMV [15]. What remains unknown is whether CMV status affects NK cell mobilization during exercise based on their expression of chemokine receptors, and thus potential tissue localization/destination. Although the exercise-responsiveness of NK cells in healthy adults and in those with infection history, such as CMV seropositive adults, is well documented, less is known about the impact of acute exercise and CMV on tissue-resident subsets of ILCs.

Helper ILCs are more likely to be tissue-resident rather than being present in the circulating blood (less than 1% of lymphocytes) [18]; however, NK cells (10%–15% of lymphocytes) circulate in the peripheral blood, protecting against virus-infected or malignant cells [16]. Given the clinical significance of tissue-resident ILCs, and the known stress-responsiveness of NK cells, understanding the effects of exercise on the mobilization of other ILC subsets in the peripheral blood is crucial to elucidating the mechanisms behind the health benefits of regular exercise. In this context, in this study we aimed to investigate the exercise responsiveness of different ILC subsets, along with their potential tissue of origin, using chemokine receptor expression as a marker of origin/destination. Furthermore, we aimed to investigate whether prior CMV infection influenced the exercise-induced mobilization of ILCs.

## 2. Materials and Methods

### 2.1. Participants

A total of 22 healthy (CMV+ n = 10; CMV− *n* = 12) men and women (mean ± SD: male *n* = 13, age 29 ± 5 years; weight 85.4 ± 16.1 kg; height 178.6 ± 7.7 cm; female *n* = 9, age 24 ± 4 years, weight 66.3 ± 6.4 kg, height 165.0 ± 6.7 cm) participated in this study. All participants provided written informed consent prior to their participation, which was approved by the Louisiana State University’s Institutional Review Board. Any individual with metabolic, inflammatory, or cardiovascular disease was excluded from this study. The participants who met the inclusion/exclusion criteria completed 2 study visits.

### 2.2. Experimental Design

The experimental protocol consisted of 2 study visits conducted within 2 weeks (minimum of 2 days). The baseline visit (V1) included the informed consent; self-reported health and medical history; and measurements of height (cm), weight (kg), and peak oxygen uptake test (VO_2Peak_). Before the second visit (V2), participants were asked to refrain from vigorous exercise and food and drinks other than water for 12 h. During V2, participants completed 30 min of exercise at a moderate/vigorous intensity on a cycle ergometer, with venous blood samples drawn immediately before and after exercise.

### 2.3. Experimental Procedures

#### 2.3.1. Height, Weight, PARQ, Medical and Health History Form (V1)

At V1, the participants completed the Physical Activity Readiness Questionnaire (PAR-Q) and a self-reported medical and health history. After consent was obtained, physical measurements, including height and weight, were measured.

#### 2.3.2. VO_2Peak_ Test (V1)

A staged, incremental exercise protocol on a stationary cycle ergometer (Racermate, Inc., Seattle, WA, USA) was used to determine peak oxygen uptake (VO_2Peak_). Respiratory gases (V_E_, VO_2_, VCO_2_) were collected and analyzed throughout the exercise test using a calibrated metabolic cart (ParvoMedics, Inc., Sandy, UT, USA). For all participants, the initial workload was set at 50 W for 3 min, and participants were instructed to pedal at at least 55 revolutions per minute (rpm). After the initial 3-min stage, the workload was increased by 25 W every 3 min until the respiratory exchange ratio (RER) reached 1.05. After RER reached 1.05, the workload was increased by 25 W every 2 min until either the participant could not maintain 55 rpm or reached volitional exhaustion. Heart rate (HR) was continuously monitored throughout the exercise protocol (Polar T31 coded transmitter, Polar Electro Oy, Kempele, Finland), and ratings of perceived exertion (RPEs) were recorded, corresponding to the last 30 s of each stage using the Borg scale [19].

#### 2.3.3. Sub-Maximal Exercise Protocol (Visit 2)

During V2, participants performed a 30-min bout of exercise on the same cycle ergometer used in the VO_2max_ test. The intensity of the exercise was based on the VO_2peak_ data and prescribed at a workload to elicit a VO2 corresponding to 65%–75% of peak VO_2_. Prior to starting the submaximal exercise bout, participants were allowed to warm up on the cycle ergometer for 10 min at 50–75W. Respiratory gases, HR, and RPE were monitored continuously and used to adjust the resistance based on biofeedback and ensure that participants remained within 65%–75% of the VO_2peak_ range.

### 2.4. Outcome Assessments

#### 2.4.1. Cell Isolation and Antibody Preparation

Plasma and serum samples (EDTA and SST vacutainers, BD and Co, Franklin Lakes, NJ, respectively) were collected before and after the submaximal exercise bout and processed immediately after the 30-min exercise protocol. A serum sample was centrifugated and frozen at −80 °C until CMV analysis. Complete blood counts were determined using an automated hematology analyzer (Sysmex XN-330, Sysmex Co., Kobe, Japan) in duplicate. Immediately after cell counting, peripheral blood mononuclear cells (PBMC) were isolated by means of density gradient centrifugation (Ficoll-Paque; GE Healthcare Bio-Sciences AB, Uppsala, Sweden), as described previously [20]. Whole blood was diluted with an equal volume of phosphate buffered saline (PBS), and 6 mL of the diluted blood was gently layered on top of 3 mL of density gradient, and then centrifuged for 30 min at 400× *g*. After centrifugation, buffy coats were harvested with a disposable transfer pipette, transferred into new tube, and washed twice in PBS. Aliquots of 1 × 10^6^ PBMCs were incubated in the dark for 45 min with pre-diluted monoclonal antibodies (mAbs) in a four-color direct immunofluorescence assay. The following mAbs were used in this study: FITC-conjugated anti-CD3 (clone OKT3), anti-CD-56 (clone TULY56), anti-lineage-cocktail (CD3, CD14, CD19, CD20), PE-conjugated anti-CD56 (NCAM CMSSB), anti-CD62L (L-selectin clone DREG56), anti-CD294 (CRTH2 clone BM16), PE-Cy7-conjugated anti- anti-CD197 (CCR7 clone 3D12), anti-CX3CR1 (clone 2A9-1), anti-CD186 (CXCR6 clone K041E5), anti-CD196 (CCR6 clone R6H1), anti-CD183(CXCR3 clone CEW33D), anti-CD185 (CXCR5 clone MU5UBEE), PE-Cy5.5 anti-CD56 (clone CMSSB), anti-CD127 (eBio clone RDR5), APC-conjugated anti-CD184 (CXCR4), anti-CD11a/CD18 (LFA-1 clone m24), anti-CD194 (CCR4 clone D8SEE, anti-CD117(c-Kit clone 104D2), anti-S1P5/EDG-8 (clone 282503), anti-CD181 (CXCR1 clone 8F1), and polyclonal APC-conjugated anti-ST2 goat IgG (IL-1R4) were used. APC-conjugated-ST2 goat IgG (IL-1R4) and APC-conjugated anti-S1P5/EDG-8 were purchased from R&D Systems (Minneapolis, MN, USA) and APC-conjugated anti-CD181 was purchased from Miltenyi Biotec (Bergisch Gladbach, Germany). The remaining antibodies were purchased from eBioscience (San Diego, CA, USA). A description of the function of the chemokine receptors and the ILC phenotype fluorochrome-conjugated antibodies used is presented in Table 1.

#### 2.4.2. Flow Cytometry Analysis

After antibody labeling, changes in ILC phenotypes were assessed on a BD Accuri C6 flow cytometer, equipped with a blue laser emitting light at a fixed wavelength of 488 nm and a red laser emitting light at a fixed wavelength of 640 nm (Accuri, Ann Arbor, MI, USA). Fluorescence emissions from the blue laser were collected via a 533/30 band-pass filter in detector FL1, a 585/40 band-pass filter in detector FL2, and a 670 long-pass filter in detector FL3. Emissions from the red laser were collected via a 675/25 filter and detected in FL4. Data analyses were performed using FCS Express (FCS Express Version 7.0, De Novo Software, Pasadena, CA, USA). The percentage of positive cells and the levels of surface chemokine expression amongst expressing subsets were recorded using median fluorescence intensity (MFI). Total cell numbers of lymphocyte subsets were determined by multiplying the percentage of each population expressing the markers of interest by the total lymphocyte count from the hematology analyzer (Appendix A).

#### 2.4.3. Determination of CMV Serostatus

All serum samples were analyzed in duplicate for their anti-CMV IgG titer via commercially available quantitative ELISA according to the manufacturer’s instructions (SERION ELISA classic Cytomegalovirus IgG/IgM; Institut Virion/Serion, Würzburg, Germany). Results were determined using a 96-well microplate reader (Molecular Devices, Sunnyvale, CA, USA).

### 2.5. Statistical Analysis

All statistical analyses were performed using JMPro 14 (SAS Inc., Cary, NC, USA). Mixed models were used to evaluate the effects of CMV status on the acute exercise response of the mobilization of ILC phenotypes and chemokine receptors on NK cells. The models included the parameters to estimate the main effect of time (pre- and post-exercise), the main effect of CMV (CMV- and CMV+), and their interaction. In addition, random effects, representing the within- and between-subject error terms, were also included in the model. The differences in cell phenotypes and numbers in response to exercise due to CMV serostatus were analyzed using the post hoc Student’s *t*-test. When main effects of CMV status and/or interactions between CMV and exercise were found, independent sample *t*-tests were used to compare the cellular ingress, defined as post-exercise cell number − pre-exercise cell number. The percentage changes in response to exercise were calculated using the mean values ((mean of post-exercise − mean of pre-exercise)/mean of pre-exercise) × 100. Similar mixed models were also used to independently evaluate the effects of sex (male and female) on the acute exercise response of the mobilization of ILC phenotypes and chemokine receptors on NK cells. Data are presented as mean ± standard deviation (SD). Statistical significance was declared at *p* < 0.05.

## 3. Results

All participants completed the 30-min sub-max cycling protocol. The physical characteristics (age, weight, height, BMI, and VO_2peak_) and exercise outcomes are presented in Table 2. No statistically significant differences were found in any physical characteristics or exercise variables during V2 between the CMV seropositive and seronegative participants (all *p* > 0.05). Of note, CMV seropositive participants attained a lower max HR and power output (W) during the VO_2peak_ test (*p* < 0.05).

### 3.1. Acute Exercise Preferentially Mobilizes NK Cell-Expressing Inflammatory Receptors and Cytotoxic Effector Functions

The total number of lymphocytes, NK cells, CD56^dim^ cells, and CD56^bright^ cells increased by 62.7% (*p* < 0.001), 204.5% (*p* < 0.001), 212.5% (*p* < 0.001), and 107.9% (*p* < 0.001), respectively, following the acute bout of cycling. The most dramatic increase in the response to exercise was observed for total NK cells’ and CD56^dim^ NK cells’ expression of CX3CR1/CXCR1 chemokine receptors, with the number of NK cells expressing CX3CR1+/CXCR1− and CX3CR1+/CXCR1+ increasing in response to the acute bout of exercise. Specifically, the number of CX3CR1+/CXCR1− receptors expressed between pre- and post-exercise increased by 218.1% amongst total NK cells (*p* < 0.001), by 227.2% amongst CD56^dim^ cells (*p* < 0.001), and by 100.3% amongst CD56^bright^ cells (*p* < 0.001) (Figure 1). Similarly, the number of NK cells expressing CX3CR1+/CXCR1+ in response to exercise increased by 209.5% (*p* < 0.001), 208.5% (*p* < 0.001), and 167.4% (*p* < 0.001) in total NK cells, CD56^dim^ cells, and CD56^bright^ NK cells, respectively (Figure 1). Furthermore, post-exercise cells expressed a greater amount of CX3CR1 receptors (CX3CR1+/CXCR1−), increasing by 13.40% in total NK cells (MFI pre: 2499.42 ± 665.49 vs. MFI post: 2824.47 ± 721.68; *p* < 0.01), and by 12.83% in CD56^dim^ cells (MFI pre: 2559.27 ± 698.85 vs. MFI post: 2885.30 ± 744.34; *p* < 0.05). A similar increase in CX3CR1 expression was also observed in cells co-expressing CXCR1+ (CX3CR1+/CXCR1+), where receptor expression increased by 11.66% in total NK cells (MFI pre: 3035.66 ± 859.79 vs. MFI post: 3389.65 ± 896.01; *p* < 0.05), by 11.97% in CD56^dim^ cells (MFI pre: 3149.92 ± 890.41 vs. MFI post: 3527 ± 929.58; *p* < 0.05), and by 11.64% in CD56^bright^ cells (MFI pre: 2390.65 ± 794.70 vs. MFI post: 2669.01 ± 1015.69; *p* < 0.001) in response to the acute bout of exercise.

The second largest exercise-induced NK cell ingress into the peripheral blood compartment was observed in homing receptors to the lymph node, CCR7/CD62L, especially in CCR7−/CD62L+ receptors. The number of total NK cells expressing CCR7−/CD62L+ increased by 153.1% (*p* < 0.001) and by 159.3% in CD56^dim^ cells (*p* < 0.001). Amongst CD56^bright^ cells, CCR7−/CD62L+-expressing cells exhibited the second largest increase in the peripheral blood post-exercise (*p* < 0.001). However, although NK cells expressing CCR7+/CD62L+ also increased in response to exercise (*p* < 0.001), the number of NK cells expressing CCR7+/CD62L− did not change from resting values (*p* > 0.05). Since CXCR3/CCR4 expression is associated with cellular migration to inflamed sites in the lymph nodes and skin, the increase in CXCR3+/CCR4− cells in response to exercise should be noted (total NK: 150.36%, *p* < 0.001; CD56^dim^: 157.19%, *p* < 0.001; CD56^bright^: 123.61 *p* < 0.001).

NK cells expressing receptors related to skin, lung, and gut mucosal immunity increased in response to exercise. Specifically, the number of cells expressing CCR6+/CCR4− increased by 142.69%, (*p* < 0.001), by 170.79% (*p* < 0.01), and by 104.2% (*p* < 0.001) in total NK cells, CD56^dim^ cells and CD56^bright^ cells, respectively. Moreover, the levels of CCR6+/CCR4+ (total NK:135.40%, *p* < 0.001; CD56^dim^: 279.78%, *p* < 0.05; CD56^bright^: 62.98%, *p* < 0.01) and CCR6-/CCR4+ (total NK: 208.51%, *p* < 0.05; CD56^dim^: 239.98%, *p* < 0.05, CD56^bright^: 80.83, *p* < 0.01) receptors were greater post-exercise.

In NK cells expressing CXCR5/S1P5 receptors, which can be found in lymphoid follicles and which exit into the blood from the bone marrow, CD56^bright^ CXCR5−/S1P5+ expression showed the largest ingress among the CD56^bright^ phenotypes (*p* < 0.05). Of the NK cells expressing the lymph node receptor CCR7 and the chemokine receptor CXCR4, CCR7+/CXCR4− cells were preferentially mobilized in the blood (*p* < 0.001), whereas CCR7+/CXCR4+ cells were the least exercise-responsive (*p* < 0.05).

Finally, the number of CXCR6+/LFA-1− NK cells increased by 167.72% (*p* < 0.001). This increase was further observed amongst CD56^dim^ cells, which increased by 179.90% following exercise (*p* < 0.001). Of note, total NK cells and CD56^dim^ cells expressing CXCR6−/LFA-1+ increased by 208.67% and 221.91%, respectively (*p* < 0.05), suggesting enhanced cytotoxic function following exercise. Regardless of CMV status, CXCR6+/LFA-1+ chemokine receptors, which are related to homing to the liver and cytotoxicity significantly increased in all NK cells (*p* < 0.01) and in both CD56^dim^ (*p* < 0.001) and CD56^bright^ (*p* < 0.01) cells (Figure 1 and Appendix A).

### 3.2. Effects of Exercise and Sex on Chemokine Receptors in NK Cells

Although no differences in the number of CD56^dim^ cells expressing CCR6+ and CCR4+ by sex were found, a higher expression of CCR6 cells was observed in CD56^dim^ cells in men following exercise (MFI men: 3448.85 ± 2628.12 vs. MFI women: 1795.12 ± 899.12; *p* < 0.05), whereas no differences were seen between sexes at rest (*p* = 0.21). No differences in the expression of the other chemokine receptors studied were observed on NK cells between sexes (*p* > 0.05).

### 3.3. CMV Infection History Increases the Mobilization of NK Cells Expressing Inflammasome and Cytotoxic Receptors

The effect of CMV serostatus on NK cell mobilization can be found in Figure 2. The number of CX3CR1+/CXCR1+ total NK cells mobilized in the blood by exercise was greater in CMV^pos^ than CMV^neg^ (*p* < 0.01), as well as amongst CD56^dim^ subsets (*p* < 0.01). There was no effect of CMV serostatus on the mobilization of CX3CR1+/CXCR1+ CD56^bright^ cells (*p* > 0.05) (Figure 2).

In addition to CMV^pos^ participants mobilizing a greater number of CX3CR1+/CXCR1+ cells in response to exercise than their seronegative counterparts, the magnitude of CX3CR1 expression on total NK cells was also greater in CMV^pos^ participants post-exercise, but not in CMV^neg^ participants (CMV^pos^ MFI pre: 3488.56 ± 879.01 vs. CMV^pos^ MFI post: 4059.6 ± 909.45; *p* < 0.05 vs. CMV^neg^ MFI pre: 2665.11 ± 671.40 vs. CMV^neg^ MFI post: 2841.51 ± 359.50; *p* > 0.05). A similarly higher CX3CR1 expression on CD56^dim^ cells was also observed at rest in CMV^pos^ participants compared to CMV^neg^ participants (MFI CMV^pos^: 3602.31 ± 899.46 vs. MFI CMV^neg^: 2779.78 ± 724.06; *p* < 0.05).

The levels of ingress of total NK cells co-expressing CXCR6 and LFA-1 were significantly different between participants with different CMV status (CMV^pos^: 2.65 ± 2.67 cell/μL vs. CMV^neg^: 0.27 ± 0.52 cell/μL; *p* < 0.05). This was observed in both CD56^dim^ cells and CD56^bright^ NK cells (CD56^dim^ CMV^pos^: 1.74 ± 1.59 cell/μL vs. CMV^neg^: 0.45 ± 0.85 cell/μL; *p* < 0.05; CD56^bright^ CMV^pos^: 0.53 ± 0.03 cell/μL vs. CMV^neg^: 0.03 ± 0.07 cell/μL; *p* < 0.05) (Figure 2).

CMV effects were observed on the number of total NK cells expressing CXCR3+/CCR4−, where CMV^neg^ participants had more pronounced increases in cell numbers following exercise (pre: CMV^pos^: 6.09 ± 3.02 cell/μL vs. post: CMV^pos^: 14.25 ± 8.20 cell/μL; *p* < 0.05; pre: CMV^neg^: 10.42 ± 5.15 cell/μL vs. post: CMV^neg^: 26.93 ± 17.08 cell/μL; *p* < 0.001). However, the number of NK cells entering the blood in response to exercise was not different between participants with differing CMV serostatus (*p* > 0.05). Similarly suppressed patterns were found in CD56^bright^ and in CD56^dim^ NK cells (*p* > 0.05).

The expression of CXCR3 was higher in CMV^pos^ participants compared to CMV^neg^ participants, especially in CD56^bright^ cells (NK cell MFI^CXCR3^ CMV^pos^: 4399.02 ± 1306.47 vs. MFI^CXCR3^ CMV^neg^: 3342.48 ± 1042.65; *p* < 0.05, CD56^bright^: MFI^CXCR3^ CMV^pos^: 4235.51 ± 1146.85 vs. MFI^CXCR3^ CMV^neg^: 3218.46 ± 924.59; *p* < 0.05).

### 3.4. The Impact of Exercise on ILCP and ILCs

The effects of exercise on the mobilization of the total number of lymphocytes and ILCs are presented in Table 3. After exercise, the number of total ILCs (tILC), ILCPs, helper-ILC1 cells, and ILC3 cells were increased by 429.9% (*p* < 0.001), 313.8% (*p* < 0.001), 450.4% (*p* < 0.001), and by 47.7% (*p* < 0.001) respectively. However, the total number of ILC2 cells decreased by 30.5% (*p* < 0.05) following exercise (Figure 3).

### 3.5. The Impact of CMV Serostatus and Response to Exercise on ILCs and Chemokine Receptors

The effect of CMV serostatus on ILC mobilization can be observed in Figure 4. Interestingly, total ILC subsets (tILC) were mobilized in the peripheral blood compartment to a greater extent in CMV^neg^ participants compared to CMV^pos^ participants (*p* < 0.05). In addition, the ingress of ILCP appeared to be suppressed in CMV^pos^ participants compared to CMV^neg^ participants following exercise (*p* < 0.05). Although CMV^neg^ exhibited a 381.82% increase from the resting state, CMV^pos^ only experienced a 205.29% increase in ILCPs post-exercise, highlighting the difference in exercise responsiveness of ILCPs depending on CMV serostatus. Although we showed that helper ILC1 and ILC3 cells were mobilized in the blood in response to acute exercise, no differences in mobilization were observed between CMV^pos^ and CMV^neg^ participants (*p* > 0.05). Since ILC2 cells were not mobilized by the acute bout of exercise, no differences in ILC2 mobilization were observed between seropositive and seronegative participants (*p* > 0.05) (Figure 4).

## 4. Discussion

Exercise-induced alterations in circulating immune cell phenotypes and functions have been well documented; however, most published research has focused on circulating T cells and NK cells [15,20]. In contrast, helper ILCs are primarily present in secondary lymphoid organs and tissues [9], suggesting that they may not be affected by physical stimuli such as acute exercise. In addition, other confounding factors have been shown to affect cellular mobilization in response to acute exercise, including psychological well-being and sex [29], along with CMV infection status [30,31]. This is the first study to examine the effects of acute exercise on the mobilization of ILC subsets, which are traditionally regarded as being tissue-resident and thus unlikely to respond to an acute stressor. In addition, we aimed to further characterize the effects of latent CMV infection and sex on exercise-induced changes in cytotoxic ILC1 cells (NK cells) expressing different chemokine receptors and non-NK-cell ILCs in response to an acute bout of sub-maximal exercise. Here we show for the first time that acute moderate-to-high-intensity exercise is a sufficient stressor to mobilize tissue-resident ILCs into the peripheral blood compartment. In addition, we show that latent CMV infection is associated with compromised ingress of total ILCs and innate lymphoid cell precursors (ILCPs) in healthy young adults, further advocating for potential detrimental indirect health effects of CMV infection later in life.

Given that ILCPs can differentiate into diverse subsets of ILCs depending on their exposure to multitudes of stimuli in local microenvironments [10], characterizing the ability of ILCs to exit tissues and to be mobilized into the periphery is essential. Moderate-to-high-intensity exercise has been proposed as a safe and potent stress-inducer for the mobilization of hematopoietic stem cells (HSCs) and has been used in a clinical setting to enhance tissue repair [32]. Exercise-induced cytokines, such as MCP-1 and CXCL12, are secreted from tissue and can transiently increase stem cell numbers following acute exercise [32]. The present data show that multi-potent ILCPs exhibit high exercise responsiveness, which could facilitate the replenishment of tissue-resident ILC pools by enhancing tissue redistribution and promoting transient exercise-induced activation. This suggests that the pool of “sedentary” tissue-resident helper ILCs could be further maintained via regular exercise [33]. As such, acute exercise could be utilized as a possible future method to augment cell frequencies for transplantation, which can be isolated and expanded further from patients/donors and thus generate other ILC subsets in vitro more rapidly and economically than current approaches [34].

Emerging evidence suggests that ILCs play a crucial role in affecting clinical outcomes in diseased populations [34,35]. For example, acute leukemia patients with high frequencies of ILCs (especially ILC3) in the blood before chemotherapy and with allogeneic HSC transplantation displayed reduced risks of developing graft-versus-host disease (GVHD) compared to patients with a lower number of ILCs, indicating that ILCs have protective effects against potential tissue damage caused by GVHD, and regulating tissue integrity [35]. Helper ILCs constantly patrol the mucosal barrier surfaces, including those of the intestine, lungs, and spleen, and interact with other immune cells in the context of cancer; therefore, ILCs are considered potential target cells for anti-tumor benefits [34]. In addition, ILC2s are enriched in mucosal barriers and contribute to allergic asthma, mediated by type 2 inflammation, which negatively regulates ILC2 cells [36]. Our data demonstrate that the number of circulating ILC2 cells decreased after acute exercise. One possible mechanism for this blunted exercise response in ILC2 cells is likely due to the increased circulating catecholamines during cycling. The increased amount of catecholamines, such as norepinephrine or epinephrine, in response to moderate-to-high-intensity exercise is well documented [1,3] and is likely to bind to β2-adrenergic receptors on the ILC2 cells, which would inhibit their activity. This promising result could further explain the known positive effects of aerobic exercise on reducing asthma symptoms [37]. We also found that CMV infection history was associated with dampened ILC mobilization following acute exercise in healthy young adults. Indeed, latent CMV infection history has dichotomous effects on tILC and ILCP mobilization. Although both NK cells and ILCs originate from common ILC precursors, the question of whether CMV infection affects ILC development or function needs to be determined.

Although an increasing body of literature shows the preferential mobilization of cytotoxic ILC1 NK cells in response to exercise, little is known about the tissue origin of these cells. As the levels of exercise-induced myokines increase, such as IL-6, IL-8, and MCP-1, which potentiate leukocyte recruitment, the ingress of cytotoxic ILC1 NK cells are increased [2]. Our study confirms that circulating cytotoxic ILC1 cells (NK cells) mobilized into the peripheral blood express chemokine receptors related to tissue inflammation, which potentially draw the cells to inflammatory sites to repair tissue in response to an acute bout of exercise. Among the chemokine receptors expressed in NK cells, CX3CR1 (fractalkine receptor) was the most expressed receptor and is a strong inflammatory mediator that controls the recruitment and activation of NK cells within inflammatory sites [23]. This chemokine receptor is expressed throughout a variety of organs and tissues, including the brain, lung, and liver. The basal expression of CX3CR1 is quickly increased during inflammatory responses [23,38], where it augments the cytotoxic function against target cells and enhances NK cell adhesion and activation [23,38,39]. Taken together, this suggests that acute exercise induces NK cells, priming for inflammatory responses, into circulation, as various researchers have suggested in regards to the T-cell exercise response [30,31].

The present data also demonstrate that latent CMV infections have dichotomous effects on CXCR3+/CXCR1+ in total NK cells. An increased ingress of cells expressing CX3CR1+/CXCR1+ on CD56^dim^ cells was predominantly observed in CMV^pos^ participants. Past work has shown that CMV infection contributes to damages in endothelial cells by promoting the interaction between fractalkine (CX3CL), CX3CR1, and CXCR1 (receptor for IL-8), thereby enhancing the accumulation of CD56^dim^ cells in the inflamed site at rest [40,41]. As such, the preferential mobilization of CX3CR1+/CXCR1+ NK cells in response to acute exercise observed in CMV^pos^ participants compared to CMV^neg^ participants is likely explained by the rapid redistribution of “primed” NK cells from the inflamed tissues. In addition, the viral chemokine vCXCL1, which is encoded by CMV, preferentially binds to CX3CR1 and CXCR1, and mediates the migration of CD56^dim^ cells to the site of infection [42]. Thus, another possible mechanism for this preferential increase in response to exercise would be that repeated CMV reactivations increase the vCXCL1 concentration [43], which directly promotes the accumulation of exercise-responsive CX3CR1+/CXCR1+ NK cells.

A previous study showed that high-intensity exercise enhances the density of LFA-1 expression in lymphocytes [2]; however, its impact on CMV serostatus and sex is less understood. The increase in LFA-1+ expression following acute exercise found in this study is consistent with previous studies [2,5]. Moreover, CMV exerts dichotomous effects on CXCR6+/LFA-1+ expression on CD56^dim^ and CD56^bright^ cells. The exercise-induced mobilization of CD56^dim^ cells with cytotoxic effector functions against malignant or target cells suggests that exercise further enhance immune surveillance and provides protection against tumor development [8,44]. Given that hepatic CD56^bright^ cells expressing CXCR6 are relatively abundant in the liver rather than in circulation at rest [45] and can generate pools of memory-like NK cells [46], it could be hypothesized that the greater exercise-induced mobilization of CXCR6+/LFA-1+ CD56^bright^ NK cells observed in CMV^pos^ participants reflects an inflated CMV-specific memory-like CD56^bright^ cell pool that is mobilized into the blood from the liver in response to acute physiological stress.

The present data also revealed significant increases in CCR6+/CCR4− and CCR6-/CCR4+ receptors on CD56^dim^ cells following acute exercise. Given that CCR6 and CCR4 are abundantly expressed in mucosal barriers, including the skin and lungs [27], our data suggest that an acute bout of moderate-to-high-intensity exercise induces cellular ingress from mucosal tissues into the blood. In combination with the increases in LFA-1 expression, these data further advocate for the beneficial role of exercise in immune cell redeployment into peripheral tissue with primed functional receptor signaling, which could enhance immune surveillance rather than blunting immune competency [47].

Our study also highlights the exercise responsiveness of CXCR5−/S1P5+ CD56^bright^ NK cells. S1P5 has been reported to be a critical chemokine receptor that allows mature NK cells to migrate into inflamed organs from the bone marrow [21]. However, previous reports showed a preferential expression of S1P5 on CD56^dim^ cells rather than CD56^bright^ cells at rest, which is in turn associated with higher circulating CD56^dim^ differentiation levels [39,48]. In spite of these results, an increase in the cellular mobilization of CD56^bright^ rather than CD56^dim^ cells expressing S1P5+ is found in response to exercise, suggesting that exercise may in fact promote the mobilization of CD56^bright^ cells into inflamed peripheral tissue to potentiate efficient cytokine production [2,49].

This study is not without limitations. The phenotypic characterization of ILC subtypes via flow cytometry can be challenging and has historically involved the use of extensive cell surface and intracellular markers. Due to technical limitations, we opted for the use of lineage antibody cocktails to identify the effects of exercise and CMV seropositivity on the different ILC subsets. However, as our understanding of ILC phenotypes expands, future studies should confirm these findings using a more comprehensive phenotypic panel.

ILCs have been studied for the past decade [9,12]; however, this is the first study to investigate the effects of acute exercise on “tissue-resident” ILCs. An increasing body of literature has established the effects of aging on the function of conventional T cells and their phenotype [20], yet little is known on the exercise effects on ILCs. Similar to unconventional T cells (i.e., MAIT, NKT, and γδ T cells), ILCs can be easily found near the barrier sites such as the skin, lung, and mucosal tissues, rather than in circulation, playing an important role in cancer, asthma, and inflammatory bowel disease [50]. Here, we show for the first time that although these cells can be mobilized in the peripheral blood compartment, their exercise responsiveness depends on CMV infection history. Since little is known on the exact function of tissue-resident ILCs when present in the circulation and their effects on other circulating immune cells, future studies should focus on the effect of short-term and long-term exercise on these ILCs and how it relates to human health and disease. Moreover, further research is needed to establish the impact of herpesviruses such as CMV and EBV on ILC function across the lifespan and healthspan.

## 5. Conclusions

In summary, this study showed that a single bout of moderate-to-high-intensity exercise is a powerful immune cell stressor, sufficient to mobilize traditionally tissue-resident ILCs into the blood. Furthermore, we show that a latent CMV infection history reduces the ingress of tILCs and ILCPs, thus reducing the overall capacity to replenish other ILCs after the completion of the exercise bout. Moreover, we confirmed that acute exercise mobilizes NK cells, especially amongst the CD56^dim^ subset, with a high expression of cytotoxic function and inflammatory receptors, indicating immune surveillance rather than immune copromotion in healthy young adults.

## Figures and Tables

**Figure 1 viruses-13-01535-f001:**
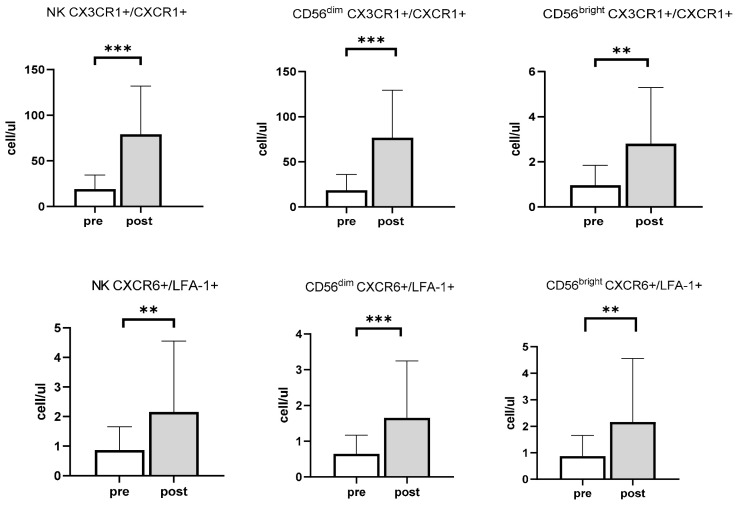
The effects of acute exercise on the mobilization of total NK cells, CD56^dim^ cells and CD56^bright^ cells expressing CX3CR1+/CXCR1+ and CCR6+/LFA-1+ in healthy young adults (*n* = 22). Pre-exercise values are represented in white, whereas post-exercise values are represented in light gray. ** Significant differences between pre- and post-exercise, *p* < 0.01. *** Significant differences between pre- and post-exercise *p* < 0.001. Data are mean ± SD.

**Figure 2 viruses-13-01535-f002:**
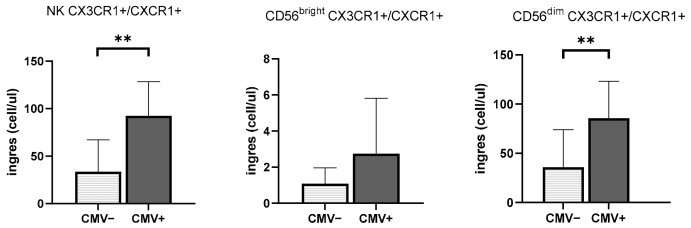
The effect of CMV seropositivity on the mobilization of total NK cells, CD56^dim^ cells, and CD56^bright^ cells expressing CX3CR1+/CXCR1+ and CCR6+/LFA-1+ in healthy young adults (*n* = 22). Changes in NK cells between resting and post-exercise are represented in light gray for CMV^neg^ participants and in dark gray for CMV^pos^ participants. * Significant differences between CMV^pos^ and CMV^neg^ participants, *p* < 0.05. ** Significant differences between CMV^pos^ and CMV^neg^ participants, *p* < 0.01.

**Figure 3 viruses-13-01535-f003:**
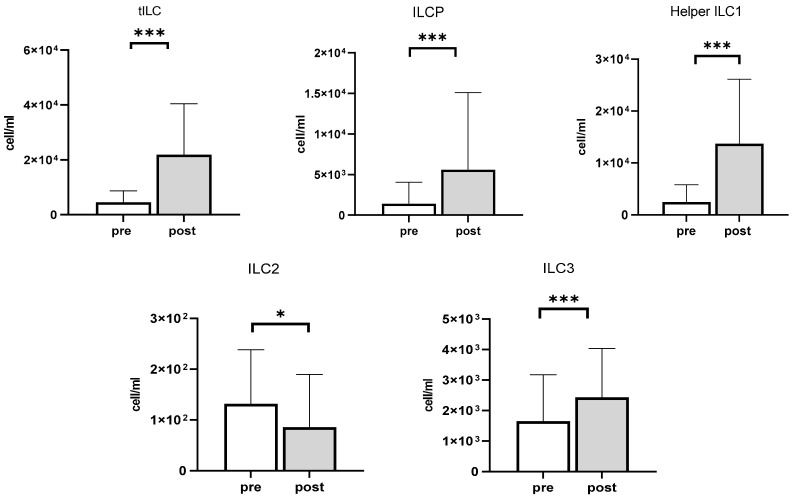
The effects of acute exercise on the mobilization of discrete ILC subsets in healthy young adults (*n* = 22). Pre-exercise values are represented in white, whereas post-exercise values are represented in light gray. * Significant differences between pre- and post-exercise *p* < 0.05, *p* < 0.01. *** Significant differences between pre- and post-exercise, *p* < 0.001. Data are mean ± SD.

**Figure 4 viruses-13-01535-f004:**
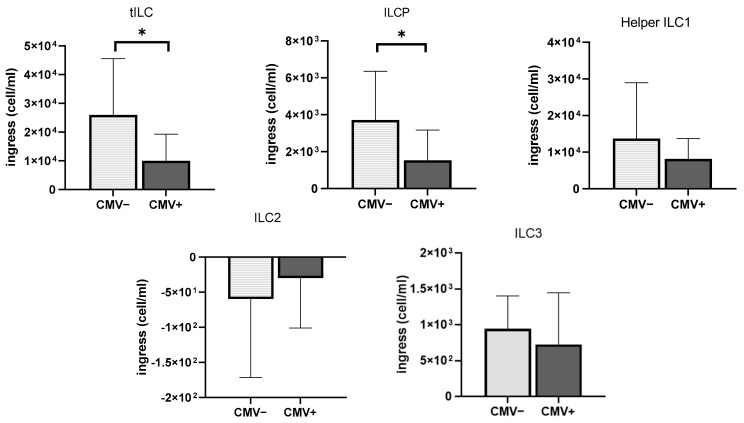
The effect of CMV seropositivity on the mobilization of the different ILC subsets in healthy young adults (n = 22). Changes in ILC counts between resting and post-exercise are represented in light gray for CMV^neg^ participants and in dark gray for CMV^pos^ participants. * Significant differences between CMV^pos^ and CMV^neg^ participants, *p* < 0.05. Data are mean ± SD.

**Table 1 viruses-13-01535-t001:** Cell surface marker combinations used to identify the different ILC phenotypes and chemokine receptors.

Cell Type	Phenotype	References
Total ILC	Lin-CD56-CD127+	[10]
ILCP	Lin-CD56-CD117+CD127+	[10]
Helper ILC1	Lin-CD56-CD127+ CD117-	[9]
ILC2	Lin-CD56-CRTH2+ ST2+	[9,13]
ILC3	Lin-CD56-CXCR5-S1P5+	[11,14]
**Chemokine Receptors**	**Roles**	**References**
CCR7+/CXCR4+	Homing receptor into lymph organs/ Retention of cells in bone marrow	[21,22]
CXCR4+/CD62L+	Retention of cells in bone marrow/ Regulation of homing into lymph nodes	[21,22]
CXCR3+/CXCR1+	Homing to inflamed tissues/ Homing to inflamed lymph node and airways	[23,24,25]
CXCR6+/LFA-1+	Homing into liver/ Regulation of activation and cytotoxicity	[2,26]
CCR6+/CCR4+	Homing to gut, mucosal tissues/ Homing to skin and lung	[25,27]
CXCR3+/CCR4+	Recruitment, activation of immune cell within inflamed tissue/ Homing to skin and lung	[25,28]
CXCR5+/S1P5+	Required for homing into lymphoid follicle/ Required to exit from bone marrow into inflamed tissue	[21,25]

**Table 2 viruses-13-01535-t002:** Physical characteristics and exercise performance of the participants (*n* = 22).

	All subjects (*n* = 22)	CMV- (*n* = 12)	CMV+ (*n* = 10)
% female	41% (*n* = 9)	25% (*n* = 3)	60% (*n* = 6)
Age (yr)	27.18 ± 5.34	28.08 ± 4.42	26.10 ± 6.35
Height (cm)	173.00 ± 9.87	174.45 ± 11.54	171.27 ± 7.64
Body weight (kg)	77.61 ± 16.03	83.24 ± 16.41	70.85 ± 13.33
BMI	25.79 ± 3.90	27.22 ± 3.66	24.08 ± 3.63
VO_2Peak_ (mL/kg/min)	36.51 ± 6.20	38.16 ± 5.82	34.53 ± 6.34
Max power (Watts)	216.25 ± 37.94	240.00 ± 29.44	192.50 ± 30.30 **
Max HR (beats/min)	186.05 ± 13.37	192.09 ± 9.30	179.40 ± 14.40 *
Sub-max exercise measures			
Mean VO_2_ (mL/kg/min)	24.95 ± 4.56	25.43 ± 4.83	24.41 ± 4.44
Mean VO_2_ (% max)	69.19 ± 14.1	65.74 ± 5.23	72.98 ± 19.51
Mean HR (beats/min)	156.76 ± 16.26	156.50 ± 12.04	157.05 ± 20.64
Mean HR (% max)	84.30 ± 6.74	81.46 ± 4.67	87.42 ± 7.49

Significant difference between pre and post are indicated by * *p* < 0.05, and ** *p* < 0.01.

**Table 3 viruses-13-01535-t003:** Total number of peripheral blood lymphocytes, NK cells (cells/μL), and ILCs (cells/mL) in response to acute exercise (*n* = 22).

	Pre	Post	Ingress Cells
Lymphocytes	1963.41 ± 389.71	3194.09 ± 907.64 ***	+1230.68
NK cell (cells/μL)	195.91 ± 84.40	596.60 ± 207.24 ***	+400.69
CD56^dim^ (cells/μL)	181.33 ± 84.36	566.58 ± 206.43 ***	+385.25
CD56^bright^ (cells/μL)	14.45 ± 6.20	30.04 ± 17.36 ***	+15.59
Total ILC (cell/mL)	4377.90 ± 4096.87	23,200.18 ± 18,271.89 ***	+18,822.28
ILCP (cell/mL)	871.06 ± 853.31	3604.12 ± 2739.67 ***	+2733.06
Helper ILC1 (cell/mL)	2492.44 ± 3307.85	13,718.13 ± 12,413.57 ***	+11,225.69
ILC2 (cell/mL)	131.73 ± 106.70	85.71 ± 103.94 *	‒37.72
ILC3 (cell/mL)	1648.66 ± 1523.97	2435.39 ± 1597.79 ***	+786.72

Significant differences between “pre” and “post” are indicated by * *p* < 0.05, and *** *p* < 0.001.

## Data Availability

The data presented in this study are available on request from the corresponding author.

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
