# Peer review of "Cytomegalovirus Infection Impairs the Mobilization of Tissue-Resident Innate Lymphoid Cells into the Peripheral Blood Compartment in Response to Acute Exercise"

_viruses, 2021, doi:10.3390/v13081535_

Round 1

Reviewer 1 Report

This paper investigates a small cohort of healthy people differing in their CMV serostatus concerning the mobilization of tissue-resident innate lymphoid cells into the peripheral blood as induced by acute exercise. Essentially, the study finds CMV serostatus affects the mobilization of tissue-resident NK cells in a negative manner indicating that CMV infections might have indirect deleterious effects in healthy adults. The study is well performed, appropriate statistics have been utilized to analyze the data. The paper itself in clearly written, my only recommendation for further improvement  relates to the fact that the results text contains many data (e.g. page 6, line 216, here all pre- and post-exercise data are listed in the text) that impede the readability of the text. These data should better be shown as a table or as supplemental material (given the fact that the data are also presented as graphs).

Author Response

We thank the reviewer for their insightful and constructive comments on our paper.

We agree with the reviewer, the initial version of our manuscript contained a lot of unnecessary data, and we have amended it. In an attempt to increase transparency, we aimed to provide as much data as possible to the reader, however we recognize that it negatively impacted the readability of the result section. We have included the data in supplemental table 1.

In addition, we have improved the grammar and spelling of the publication.

Reviewer 2 Report

In the study by Cho et al, the impact of cytomegalovirus infection on the mobilization of ILCs into the peripheral blood in response to acute exercise was assessed. While the impact of exercise and CMV infection on the NK cell compartment was assessed previously in detail, here the authors provide novel data on the expression of chemokine receptors on mobilized NK cells. In addition, it is demonstrated that a high-intensity exercise is associated with an influx of tissue-resident ILCs and that CMV seropositivity is associated with impaired exercise-response of total ILCs.  

While this data is interesting and potentially relevant, my major concern is on the technical side. As the authors correctly state, only NK cells are abundant in blood, while other ILC populations are rare in the blood, i.e. approximately 0.1-0.3% of total lymphocytes. Therefore, strict flow cytometry analysis is required to select and properly identify non-NK ILCs. Representative histograms/plots should be shown for the analyses all over the manuscript and gating. It should be stated if live cells were gated by use of live/dead stains. Major markers were used for the selection strategy of distinct ILC populations, however, in the cited literature more markers were used to distinguish these cell types. Therefore, more data should be included to be sure that the presented data efficiently identifies indicated populations of non-NK ILCs. 

Author Response

We thank the reviewer for their insightful and constructive comments on our pape

We agree with the important point raised by the reviewer. Identifying small subpopulations of immune cells in circulation can be challenging, so it requires optimal gating strategies and flow cytometry analysis. We have included our gating strategy as a supplementary material (supplementary Figure S1). We acknowledge that previous studies have used a more comprehensive antibody panel to characterize non-NK ILCs. Unfortunately however, our analysis was limited to 4 colors due to instrument-associated technical limitations. Since we included a lineage antibody cocktail in our analysis, we are confident that the data presented in this publication reflect exercise-induced changes in ILC numbers and phenotypes, however we have included this limitation to the manuscript. Similarly, we did not use a LIVE/DEAD stain to distinguish from alive and dead immune cells in our analysis. In the present study, ILCs were isolated, stained and analyzed via flow cytometry within 2 hours of sample collection. In our experience, cellular death is extremely minimal in freshly isolated samples, and the use of LIVE/DEAD stains is not required, unlike when working with previously cryopreserved cells.

In addition, we have improved the grammar and spelling of the publication.

Round 2

Reviewer 2 Report

The manuscript in the present form is suitable for publication. The authors have addressed my major concern by discussing the problems of the selection of ILC types by flow cytometry in the discussion section.